# One-pot aminobenzylation of aldehydes with toluenes

Zhiting Wang[1], Zhipeng Zheng[2], Xinyu Xu[1], Jianyou Mao [1] & Patrick J. Walsh [1,2]

Amines are fundamental motifs in bioactive natural products and pharmaceuticals. Using simple toluene derivatives, a one-pot aminobenzylation of aldehydes is introduced that provides rapid access to amines. Simply combining benzaldehydes, toluenes, $NaN(SiMe_3)_2$, and additive $Cs(O_2CCF_3)$ (0.35 equiv.) generates a diverse array of 1,2-diarylethylamine derivatives (36 examples, 56–98% yield). Furthermore, suitably functionalized 1,2-diarylethylamines were transformed into 2-aryl-substituted indoline derivatives via Buchwald–Hartwig amination. It is proposed that the successful deprotonation of toluene by $MN(SiMe_3)_2$ is facilitated by cation–$\pi$ interactions between the arene and the group(I) cation that acidify the benzylic C–Hs.

[1] Institute of Advanced Synthesis, School of Chemistry and Molecular Engineering, Jiangsu National Synergetic Innovation Center for Advanced Materials, Nanjing Tech University, 30 South Puzhu Road, 211816 Nanjing, China. [2] Roy and Diana Vagelos Laboratories, Penn/Merck Laboratory for High-Throughput Experimentation, Department of Chemistry, University of Pennsylvania, 231 South 34th Street, Philadelphia, PA 19104, USA. Correspondence and requests for materials should be addressed to J.M. (email: ias_jymao@njtech.edu.cn) or to P.J.W. (email: pwalsh@sas.upenn.edu)

oluene and xylenes are large volume, inexpensive commodity chemicals commonly used as solvents on industrial scale. As such, there are no better feedstocks for the preparation of more elaborate, high-value organic molecules with applications in pharmaceutical sciences, agrochemicals, and materials chemistry[1–3]. To fully exploit these feedstocks, efficient and economical methods for the selective functionalization of the benzylic C–Hs are required. Recent advances along these lines have been considerable, although many rely on highly reactive stoichiometric oxidants[3] or directing groups to facilitate these transformations[4,5]. Related to this strategy, Stahl[6] and Liu[7] have recently developed mild methods to generate diarylmethanes via copper-catalyzed arylations of toluene and its derivatives with arylboronic acids (Fig. 1). Palladium-promoted toluene functionalization strategies also hold promise[8,9].

We have been interested in the functionalization of very weakly acidic (p$K_a$ up to ~34) benzylic C–Hs of arenes and heteroarenes via deprotonative cross-coupling processes (DCCP). Substrates for DCCP include allyl benzenes (Fig. 2a)[10], diarylmethanes[11–14], triarylmethanes[15], and benzylic sulfoxides[16,17] among others. The success of DCCP relies partly on reversible deprotonation of the benzylic C–Hs of the pronucleophile. For unactivated toluene derivatives, however, we conjectured that the high p$K_a$ values (≈43 in DMSO[18]) of the benzylic C–Hs were far beyond the reach of MN(SiMe$_3$)$_2$ bases [M = alkali metal, p$K_a$ ≈ 26 for HN(SiMe$_3$)$_2$ in THF[19]]. Thus, to address this long-standing challenge, we[20,21] and others[22] activated the arenes with stoichiometric transition metals by forming ($\eta^6$-toluene)Cr(CO)$_3$ complexes. The benzylic C–Hs of ($\eta^6$-toluene)Cr(CO)$_3$ exhibit increased acidity and are reversibly deprotonated with LiN(SiMe$_3$)$_2$, enabling functionalization (Fig. 2b). The group of Matsuzaka improved upon this approach with a ruthenium-sulfonamide-based catalyst (Fig. 2c) for in situ deprotonation of toluene and dehydrative condensation with aromatic aldehydes to generate (E)-stilbenes[23].

As a valuable complement, Schneider developed a method for functionalization of allyl benzene (p$K_a$ ≈ 34[24]) catalyzed by NaN(SiMe$_3$)$_2$ (Fig. 2d)[25]. Benzylic functionalization of more acidic alkylazaarenes (p$K_a$ ≈ 35 for 4-methyl pyridine[18]) catalyzed by KN(SiMe$_3$)$_2$ with N,N-dimethylcinnamamide by Kobayashi and co-workers also represents an advance (Fig. 2e)[26,27]. More recently, Guan reported a KN(SiMe$_3$)$_2$-catalyzed C–H bond addition of alkylpyridines to simple styrenes (Fig. 2f)[28]. During the revision process, Brønsted base-catalyzed benzylic C–H bond functionalizations of toluenes and diarylmethanes were reported by Kobayashi[29] and Guan[30], respectively. Important early contributions involved additions of 2-methyl pyridine (Fig. 2g), and Grignard reagents (Fig. 2h) to in situ generated N-(trimethylsilyl)imines were developed by Giles[31] and Hart[32,33], respectively.

The results in Fig. 2b–h, as well as our recent work on the use of cation–π interactions to direct C–H functionalization reactions[11], inspired us to wonder if cation–π interactions between toluene and earth-abundant alkali metals derived from MN(SiMe$_3$)$_2$ (M = Li, Na, K, Cs) would increase the acidity of the benzylic C–Hs sufficiently to allow reversible deprotonation under relatively mild conditions. If indeed such an equilibrium

could be established, which would undoubtedly lie very far to the side of toluene, would it be possible to trap the fleeting benzyl organometallic with an electrophile before rapid quenching with the conjugate acid of the base [HN(SiMe$_3$)$_2$]? Finally, if benzylic organometallic species could be generated from toluene and its derivatives, would it be possible to transform them into high-value added building blocks of interest to the pharmaceutical industry?

Herein, we report a successful tandem C–C and C–N bond-forming reaction for the one-pot chemoselective aminobenzylation of aldehydes with toluene derivatives (Fig. 2i). This method enables rapid access to a variety of 1,2-diphenylethylamine derivatives that are important building blocks in natural products and potent drugs and pharmaceuticals (NEDPA, NPDPA, lefetamine, ephenidine, MT-45, and PAO1, Fig. 2j)[34–36].

## Results

**Preliminary reaction optimization.** Initial screens were conducted with benzaldehyde (**1a**) in toluene (**2a**) with three different bases [LiN(SiMe$_3$)$_2$, NaN(SiMe$_3$)$_2$, and KN(SiMe$_3$)$_2$] at 110 °C for 12 h (Table 1, entries 1–3, AY = assay yield, determined by $^1$H NMR of unpurified reaction mixtures). LiN(SiMe$_3$)$_2$ failed to give the desired product **3aa** (entry 1), although it is known to react with benzaldehyde to generate aldimine[37,38]. This screen led to the identification of NaN(SiMe$_3$)$_2$ as a promising base, affording the product **3aa** in 47% assay yield (entry 2). It is known that K$^+$ forms the stronger cation–π interactions in solution in the series Li$^+$, Na$^+$, and K$^+$[39,40]; however, under our conditions the potassium amide was not as successful as NaN(SiMe$_3$)$_2$ (entry 3 vs. 2). This may be because KN(SiMe$_3$)$_2$ is less efficient in the formation of the aldimine[37]. We were also interested in examining the use of CsN(SiMe$_3$)$_2$. Unfortunately, this base is not widely available commercially like its lighter analogs[41]. It can be prepared, of course, but this would make the methods that require its use less attractive. O'Hara and co-workers found that CsN(SiMe$_3$)$_2$ could be generated by mixing NaN(SiMe$_3$)$_2$ with CsX (X = Cl, Br, or I)[40]. Furthermore, combining CsN(SiMe$_3$)$_2$ with equimolar NaN(SiMe$_3$)$_2$ led to formation of a sodium–cesium amide polymer [toluene•CsNa(N(SiMe$_3$)$_2$)$_2$]$_\infty$[39]. It is noteworthy that the toluene in this structure forms a cation–π complex with the Cs$^+$ [Cs•••toluene(centroid) = 3.339 Å]. Such cation–π interactions are often found in the structures of organometallic complexes[42–44].

On the basis of the structures in these reports, we examined a variety of cesium salts with commercially available NaN(SiMe$_3$)$_2$ in 1:1 ratio (entries 4–13). This screen led to the identification of NaN(SiMe$_3$)$_2$ and CsTFA (TFA = trifluoroacetate) as the best combination, generating **3aa** in 92% AY (entry 13). Other cesium salts either did not promote the transformation (CsF, Cs$_2$CO$_3$, CsCl, CsOAc, entries 4–7) or gave low yields of **3aa** (Cs$_2$SO$_4$, CsClO$_4$, EtCOOCs, CsBr, CsI, entries 8–12). Upon decreasing the amounts of both base and CsTFA [2 equiv. NaN(SiMe$_3$)$_2$ and 0.35 equiv. CsTFA], the AY of **3aa** remained high (95%, entry 15). Further lowering the CsTFA to 0.2 equiv, however, led to a slight decrease in assay yield (89%, entry 16). The reactivity decreased significantly when catalytic MN(SiMe$_3$)$_2$ was used (41%, entry 17,

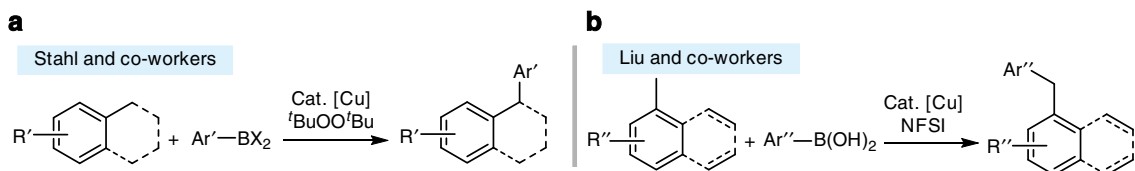

**Fig. 1** Copper-catalyzed arylations of toluene derivatives. **a** Cu-catalyzed oxidative arylation with di-*tert*-butyl peroxide. **b** Cu-catalyzed arylation with N-fluorobenzenesulfonamide (NFSI)

20 mol %). We wanted to determine if the combination of MN (SiMe$_3$)$_2$ (M = Li, K) and CsTFA could affect the reactivity under the conditions of entry 15, so we reexamined LiN(SiMe$_3$)$_2$ and KN(SiMe$_3$)$_2$ with CsTFA (35 mol%). In the presence of CsTFA, the difference between LiN(SiMe$_3$)$_2$ and NaN(SiMe$_3$)$_2$ was negligible (entry 15 vs 18). KN(SiMe$_3$)$_2$, however, still gave low assay yield (entry 19). We also examined the impact of temperature on the reactivity under the conditions of entry 15. As the temperature was decreased from 110 to 30 °C, the reactivity decreased slightly at 80 and 40 °C (entries 20 and 21). The reactivity decreased dramatically at 30 °C affording the desired product in 67% yield (entry 22). At this point, the nature of the active base and even the amount of Cs in solution remain the subject of future work. Our optimized reaction conditions for the one-pot aminobenzylation of benzaldehyde are 2 equiv. of NaN(SiMe$_3$)$_2$, 1 mL toluene, and 35 mol% CsTFA at 110 °C for 12 h.

**Scope of aldehydes.** With the optimized reaction conditions in hand, we next examined the scope of aldehydes in the aminobenzylation with toluene (Table 2). In addition to the parent benzaldehyde (**1a**), a variety of aryl and heteroaryl aldehydes were successfully employed. Benzaldehydes bearing electron-donating groups, such as 4-*t*-Bu, 4-methyl, 4-methoxy, and 4-*N*,*N*-dimethylamino, exhibited very good reactivity, producing **3ba**–**3ea** in

70–88% yield. Benzaldehydes possessing halogens are also good coupling partners even at 40 °C. 4-Fluoro-, 4-chloro-, and 4-bromobenzaldehydes afforded the corresponding products in 90%, 98%, and 83% yield, respectively (**3fa–3ha**). Likewise, 2-bromo (**3ia**, 95%) and 2-chloro (**3ja**, 98%) benzaldehydes were very good substrates. Of course, these products could potentially be further functionalized through cross-coupling reactions. Substrates with extended π-systems, such as 1- and 2-naphthyl aldehydes, furnished products in 92–94% yield (**3ka** and **3la**).

In general, benzaldehyde derivatives bearing additional functional groups and heteroatoms were well tolerated. Nitriles are known to undergo nucleophilic additions with organometallic reagents[45]. Under our aminobenzylation conditions, however, 4-cyano benzaldehyde afforded the desired product (**3ma**) in 70% yield with high chemoselectivity. Considering that, fluorinated compounds are extremely important in medicinal chemistry[46], we examined fluorinated benzaldehydes. Both 4-trifluoromethyl- and 4-trifluoromethoxy benzaldehydes were excellent substrates, affording **3na** and **3oa** in 97% and 90% yield, respectively. Benzaldehydes containing 4-SMe, 4-Ph, and 4-OPh groups rendered the products in 88–93% yield (**3pa–3ra**). The silylether containing product (**3sa**, 77%) could be accessed via this protocol. Heterocyclic amines exhibit various bioactivities[47]. Amines containing indole, pyridine, pyrrole, and quinoline groups could be prepared with our approach, as exemplified by

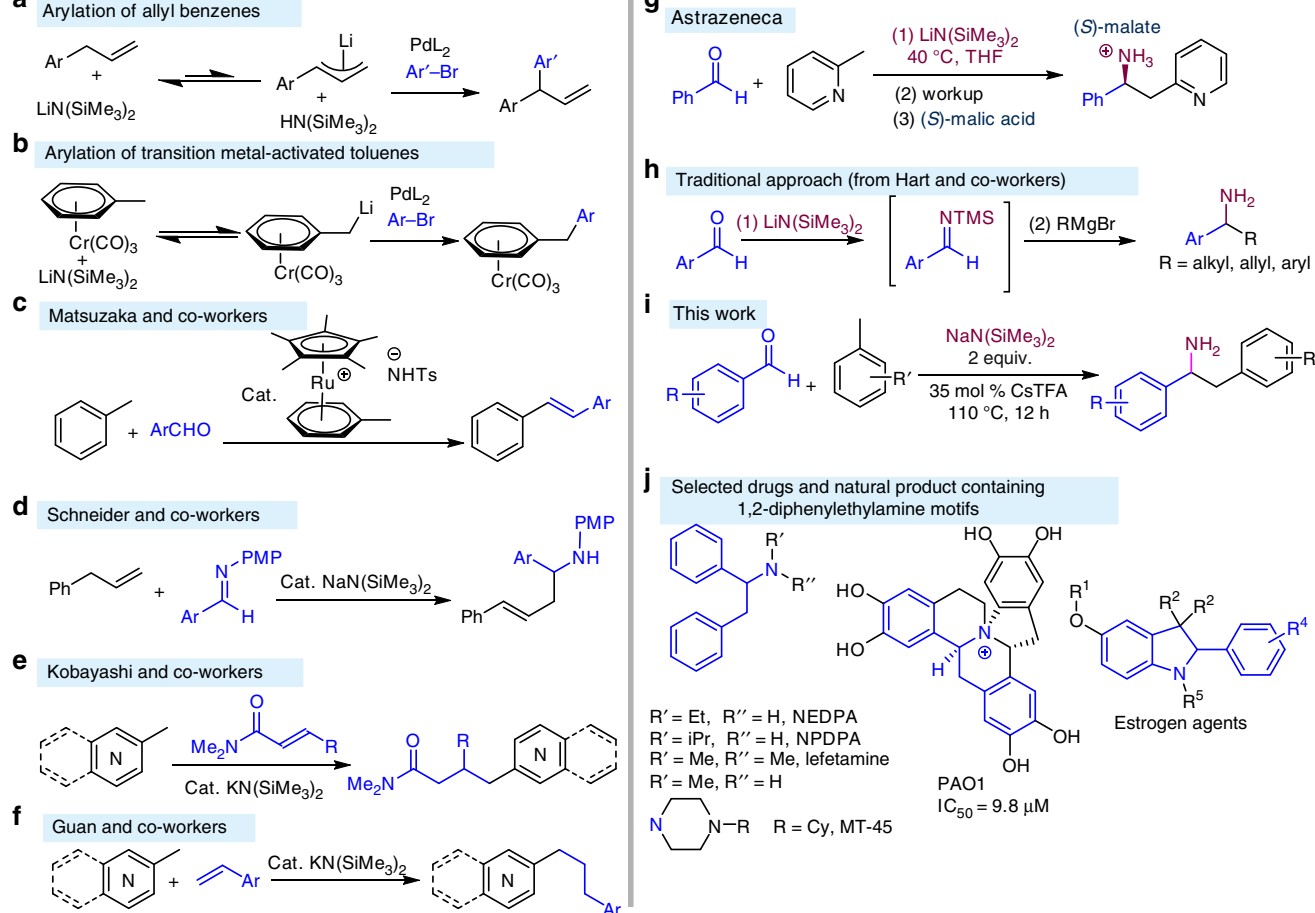

**Fig. 2** Benzylic deprotonation and related chemistry. **a** Arylation of allyl benzene. **b** Arylation of transition metal-activated toluene derivatives, **c** catalytic arene activation with a ruthenium complex. **d** Catalytic deprotonation of allyl benzene and imine addition. **e** Catalytic 1,4-addition reaction with alkylazaarenes by Kobayshi and co-workers. **f** Catalytic addition of benzylic C–Hs to styrenes by Guan and co-workers. **g** Related chemistry with more acidic 2-methyl pyridine. **h** Traditional approach from Hart and co-workers. **i** Aminobenzylation of aldehydes (this work). **j** 1,2-Diphenylethylamine-based drugs and natural products

the generation of **3ta–3xa** in 66–97% yield. Cinnamaldehyde was a competent partner under our conditions, as exemplified by the synthesis of allylic amine **3ya** in 56% yield.

**Scope of toluene derivatives**. Next, the substrate scope of toluene derivatives was examined in the aminobenzylation of benzaldehyde (**1a**) (Table 3). Toluenes bearing electron-donating groups, such as 4-$^i$Pr (**2b**), 4-OMe (**2c**), and 2-OMe (**2d**), provided the corresponding products in 78%, 66%, and 77% yield, respectively. It is noteworthy that the methyl of *p*-cymene (**2b**) undergoes reaction with high chemoselectivity. 4-Chlorotoluene (**2e**) exhibited reduced reactivity, furnishing **3ae** in 66% yield at 40 °C. In contrast, 2-chloro- and 2-bromotoluenes exhibited good reactivity, giving the desired products (**3af** and **3ag**) in 86% and 85% yield, respectively. For polymethyl-substituted toluenes (**2h–2k**), excellent chemoselectivity was observed, affording the products (**3ah–3ak**) in 81–96% yield. It is noteworthy that mesitylene was an outstanding substrate (96% yield, **3ak**). π-Extended 1-methylnaphthalene was also a good substrate, affording **3al** in 88% yield.

**Reaction pathway**. Based on the results above, we propose a reaction pathway for this one-pot aminobenzylation process. First, the NaN(SiMe$_3$)$_2$ reacts with the aldehyde to form the intermediate adduct **A**[48], followed by an aza-Peterson olefination to afford the *N*-(trimethylsilyl)imine **B**, which was not isolated but reacted directly in this one-pot process. In the presence of NaN(SiMe$_3$)$_2$ and CsTFA, the toluene derivative was reversibly deprotonated to generate an $\eta^1$- or $\eta^3$-bound metal complex **C**

and **C′**[49]. The deprotonated toluene derivative then attacks the in-situ-generated aldimine **B** to give the aminobenzylated product **3** after workup (Fig. 3).

**Further transformations**. For a method to be useful, it must be scalable. To test the scalability of the aminobenzylation, 5 mmol of benzaldehyde (0.53 g) was reacted with 2-bromotoluene (**2g**) (Fig. 4a). An 81% yield of **3ag** was obtained. Additionally, a column-free process for direct synthesis of hydrochloride salt of **3aa** was explored. Under the optimized conditions, the salt **3′aa** was obtained in 78% yield (Fig. 4b). To further demonstrate the synthetic potential of the aminobenzylation, the product derived from 2-bromotoluene, NaN(SiMe$_3$)$_2$, and benzaldehydes **1a**, **1n**, **1q**, and **1v** were readily converted into valuable 2-aryl-substituted indoline derivatives using a Buchwald–Hartwig amination in 81–90% yield (Fig. 4c)[50]. *N*-Substituted 2-arylindoline derivatives are used to treat estrogen-deficiency diseases[51]. Furthermore, the parent 1,2-diphenylethylamine was easily converted to a diverse array of biologically active compounds (Fig. 4d)[52].

**Discussion**

We have advanced a general method for the activation and functionalization of inexpensive toluene feedstocks at the benzylic position via a one-pot aminobenzylation of aldehydes. The reaction takes place without added transition metal catalysts and does not employ preformed main group organometallic reagents. By employing readily available benzaldehydes, commodity toluene derivatives, NaN(SiMe$_3$)$_2$, and substoichiometric Cs (TFA) a diverse array of valuable and biologically active 1,2-

---

**Table 1 Optimization of one-pot aminobenzylation of benzaldehyde**

| Entry | Base | Additives | Base: additives (equiv.) | AY (%)[a] |
|---|---|---|---|---|
| 1 | LiN(SiMe$_3$)$_2$ | — | 3:0 | 0 |
| 2 | NaN(SiMe$_3$)$_2$ | — | 3:0 | 47 |
| 3 | KN(SiMe$_3$)$_2$ | — | 3:0 | 35 |
| 4 | NaN(SiMe$_3$)$_2$ | CsF | 3:3 | Trace |
| 5 | NaN(SiMe$_3$)$_2$ | Cs$_2$CO$_3$ | 3:3 | 0 |
| 6 | NaN(SiMe$_3$)$_2$ | CsCl | 3:3 | 0 |
| 7 | NaN(SiMe$_3$)$_2$ | CsOAc | 3:3 | Trace |
| 8 | NaN(SiMe$_3$)$_2$ | Cs$_2$SO$_4$ | 3:3 | 20 |
| 9 | NaN(SiMe$_3$)$_2$ | CsClO$_4$ | 3:3 | 78 |
| 10 | NaN(SiMe$_3$)$_2$ | EtCO$_2$Cs | 3:3 | 17 |
| 11 | NaN(SiMe$_3$)$_2$ | CsBr | 3:3 | 34 |
| 12 | NaN(SiMe$_3$)$_2$ | CsI | 3:3 | 53 |
| 13 | NaN(SiMe$_3$)$_2$ | CsTFA | 3:3 | 92 |
| 14 | NaN(SiMe$_3$)$_2$ | CsTFA | 2:3 | 92 |
| 15 | NaN(SiMe$_3$)$_2$ | CsTFA | 2:0.35 | 95 |
| 16 | NaN(SiMe$_3$)$_2$ | CsTFA | 2:0.2 | 89 |
| 17[b] | NaN(SiMe$_3$)$_2$ | CsTFA | 1.2:0.35 | 41 |
| 18 | LiN(SiMe$_3$)$_2$ | CsTFA | 2:0.35 | 94 |
| 19 | KN(SiMe$_3$)$_2$ | CsTFA | 2:0.35 | 50 |
| 20[c] | NaN(SiMe$_3$)$_2$ | CsTFA | 2:0.35 | 92 |
| 21[d] | NaN(SiMe$_3$)$_2$ | CsTFA | 2:0.35 | 91 |
| 22[e] | NaN(SiMe$_3$)$_2$ | CsTFA | 2:0.35 | 67 |

[a]Assay yields (AY) determined by $^1$H NMR analysis of crude reaction mixture with CH$_2$Br$_2$ as internal standard
[b]In this transformation, 1 equiv. MN(SiMe$_3$)$_2$ was needed to form aldimine; the remaining 0.2 equiv. catalyzed the reaction
[c]Reaction conducted at 80 °C
[d]Reaction conducted at 40 °C
[e]Reaction conducted at 30 °C

**Table 2 Scope of aldehydes[a]**

**3aa** (92%)  **3ba** (88%)  **3ca** (70%)  **3da** (86%)  **3ea** (79%)

**3fa** (90%)  **3ga** (98%)[†]  **3ha** (83%)[†]  **3ia** (95%)[†]  **3ja** (98%)[†]

**3ka** (94%)[†]  **3la** (92%)[†]  **3ma** (70%)  **3na** (97%)[†]  **3oa** (90%)

**3pa** (88%)  **3qa** (93%)[†]  **3ra** (92%)  **3sa** (77%)  **3ta** (80%)

**3ua** (84%)  **3va** (75%)  **3wa** (97%)[†]  **3xa** (66%)  **3ya** (56%)

[a]Isolated yields
[†]Reaction conducted at 40°C

**Table 3 Scope of toluene derivatives[a]**

**3ab** (78%)  **3ac** (66%)[†]  **3ad** (77%)  **3ae** (66%)[†,§]  **3af** (86%)  **3ag** (85%)

**3ah** (82%)  **3ai** (84%)  **3aj** (81%)  **3ak** (96%)  **3al** (88%)[§]

[a]Isolated yields
[†]Reaction conducted in 1 mL 4-methylanisole
[‡]Reaction conducted in 1 mL 4-chlorotoluene
[§]Reaction conducted at 40 °C

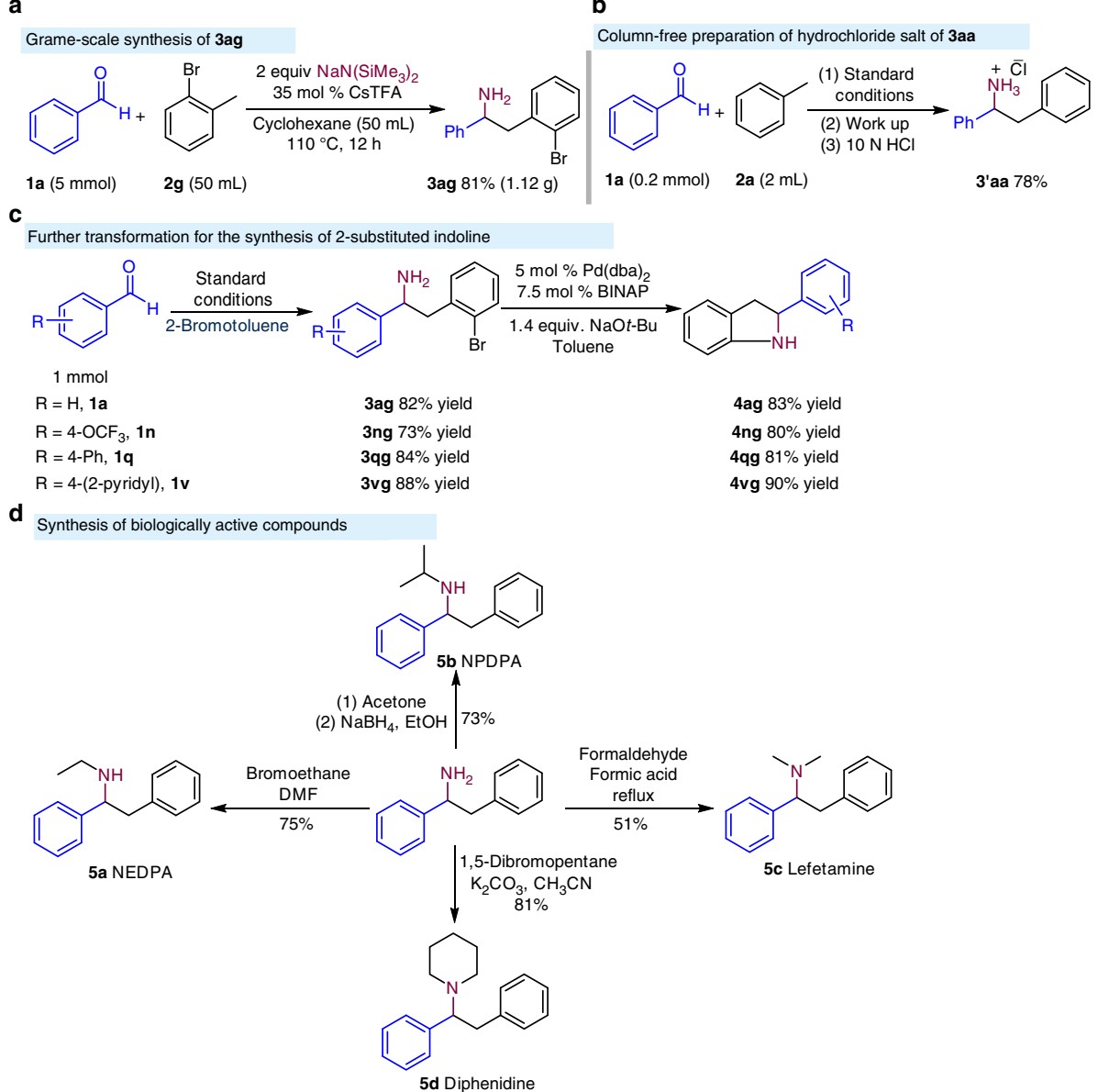

**Fig. 3** Possible reaction pathway for the aminobenzylation. In situ imine formation provides aldimine **B** while NaN(SiMe₃)₂/CsTFA-mediated benzylic deprotonation generates transient organometallic **C**. Intermediate **C** is trapped by **B** to furnish intermediate **D**. Workup then affords the desired amine product **3**

**Fig. 4** Gram scale and further transformations. **a** Scale up of aminobenzylation. **b** Column-free synthesis of the hydrochloride salt of the amine product. **c** Synthesis of indolines via Buchwald–Hartwig amination. **d** Synthesis of bioactive compounds

diarylethylamine derivatives were conveniently synthesized. The *N*-alkylated analogs of our diarylethylamines are used as opioid analgesics. Additionally, this one-pot aminobenzylation exhibits remarkable chemoselectivity and excellent functional group tolerance. Suitably substituted aldehyde aminobenzylation products were readily transformed into pharmaceutically relevant 2-aryl-substituted indoline derivatives via Buchwald–Hartwig amination.

We ascribe the success of our aminobenzylation of aldehydes to cation–π interactions between the π-electrons of the toluene derivative and Na$^+$ and/or Cs$^+$ centers. We hypothesize that this interaction acidifies the benzylic C–H bonds, facilitating deprotonation by moderate bases, MN(SiMe$_3$)$_2$[53–57]. Because of its simplicity and potential to produce valuable bioactive building blocks in a single step, we anticipate that this aminobenzylation reaction will find applications in medicinal chemistry.

## Methods
**General procedure A**. To an oven-dried microwave vial equipped with a stir bar under argon atmosphere inside a glove box was added NaN(SiMe$_3$)$_2$ (73.2 mg, 0.40 mmol), cesium trifluoroacetate (CsTFA) (17.2 mg, 0.07 mmol), and toluene (2 mL). Then, the corresponding aldehyde (0.20 mmol) was added via syringe. The microwave vial was sealed with a cap and removed from the glove box. The reaction mixture was heated to 110 °C in an oil bath and stirred for 12 h. The sealed vial was cooled to room temperature, opened to air, and then five drops of water were added. The reaction mixture was passed through a short pad of silica, washed with an additional 6 mL of ethyl acetate (3 × 2 mL), and the combined solutions were concentrated in vacuo. The crude material was loaded onto a column of silica gel for purification of the amine.

**General procedure B**. To an oven-dried microwave vial equipped with a stir bar under an argon atmosphere inside a glove box was added NaN(SiMe$_3$)$_2$ (73.2 mg, 0.40 mmol), CsTFA (17.2 mg, 0.07 mmol), the toluene derivative (1 mL), and cyclohexane (1 mL). Then, the corresponding aldehyde (0.20 mmol) was added via syringe. The microwave vial was sealed with a cap and removed from the glove box. The reaction mixture was heated to 110 °C in an oil bath and stirred for 12 h. The sealed vial was cooled to room temperature, opened to air, and then five drops of water were added. The reaction mixture was passed through a short pad of silica, washed with additional 6 mL of ethyl acetate (3 × 2 mL), and the combined solutions were concentrated in vacuo. The crude material was loaded onto a column of silica gel for purification of the amine.

**Data availability**. The authors declare that the data supporting the findings of this study are available within the article and its Supplementary Information files.

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

## Acknowledgements

The authors acknowledge the start-up grant of Nanjing Tech University (3980001601 to P.J.W. and 39837112 to J.M.) and Natural Science Foundation of Jiangsu Province, China (BK20170965) for financial support. P.J.W. thanks the US National Science Foundation (CHE-1464744). The authors are grateful for financial support by SICAM Fellowship by Jiangsu National Synergetic Innovation Center for Advanced Materials. Prof. Peng Cui (Anhui Normal University), Prof. Lili Zhao (Nanjing Tech University), and Dr. Haolin Yin (Caltech University) are thanked for helpful discussions.

## Author contributions

Z.W., X.X., Z.Z., and J.M. performed the experiments and analyzed the data. J.M. and P.J. W. conceived the project and wrote the manuscript. All authors discussed the results and commented on the manuscript.
