## [Peer Review File · Nature Communications]

Reviewer #1 (Remarks to the Author):

I already reviewed the originally submitted paper by Mao & Walsh, which is an interesting contribution in the context of formal C–H bond activation—via a supposed transition metal-free deprotonation—of challenging low-acidity substrates (toluene and derivatives; $pK_a \sim 43$).

Regarding the revised version of this manuscript, I am fairly pleased with the efforts of the authors although there are still a few points to be addressed.

The background is now (almost) complete and accurate, although a few adjustments are required in terms of order, referencing, and visibility of the chemistries displayed in Scheme 2. I believe that the benzylic C–H bond activations “D–E–F” are more accurately to be shown in a chronological order; Schneider (2017), Kobayashi (2017, plus concept article in 2018), Guan (2018); means: D & E should be swapped (likewise the corresponding refs), and the text must be adjusted accordingly. In the paper’s context (benzylic substrates), ref 26 is not required because it is not on benzylic C–H bond activation, but rather on “ordinary” catalytic enolate-type chemistry; also, this paper is cited already in the concept article – thus, please remove ref 26 (otherwise, many others would need to be added!). While it is appreciated that refs 27 and 31 (published very recently) have been included in the References, I would like to see these briefly contextualized in the text (not as a Scheme); ref 31 is not “competitive” in terms of pK_a values anyway, but ref 27 is very much “competitive” here, and it is not only catalytic but also includes the addition to alkenes as well as a preliminary asymmetric version (imine addition). Finally, please add the term “catalytic” to both Schneider’s and Guan’s works (legend of Scheme 2); these catalyses must be distinguished from the stoichiometric studies (not only in the artwork).

From a general point of view, the argumentation (in the rebuttal) about the lengthy imine preparation in Kobayashi’s work is not so relevant because their reaction is catalytic in metal amide—which should be more appreciated—and it works also with alkenes (using a tridentate amine ligand as additive); a particularly difficult transformation. Anyway, I still think that the mediator system in this work –vs. Kobayashi’s work– is somewhat different, which gives credit to the Mao & Walsh: Kobayashi used a lithium amide activated by a potassium alkoxide (Lewis base), whereas Mao & Walsh used a sodium amide and a cesium salt (π -Lewis acid), the latter of which activates the benzylic substrates. Means: the activation mode seems to be quite distinct. In this manuscript, a TON of 2 was achieved (Table 1, entry 17), which is not great but shows that catalysis should be in principle feasible.

It is fair to say that the various (other) points raised earlier by myself and by the other reviewers have been addressed satisfactorily. For instance, the distinction to the AZ work is not great but sufficient. The delivered control experiments and explanations are fine (T1 & S1); likewise, the displayed scope (T2 & T3) is acceptable; I am still not convinced regarding the utility of S4 though. The requested SI adjustments are fine as well. The displayed mechanism (S3) is straightforward; one

detail –regarding the structure of key intermediate C– must be checked/adjusted though prior to publication: is it really to be represented as a sigma-complex, or isn't it rather a pi-complex (like known for e.g. benzyl–Pd species)? Thus, please verify in the literature the eta1 vs. eta3 complexation mode for this alkali metal–benzyl intermediate. It should be known in an inorganic context. It may be best to draw both species in an equilibrium anyway (?).

Conclusion: Because of the highly challenging pKa value of the substrates, the distinct activation mode (vs. Kobayashi 2018), and the catalytic proof-of-principle (while not great yet), I would say that I can recommend publication in Nat. Commun. if the mentioned points above are addressed in full.

Reviewer #3 (Remarks to the Author):

This revised manuscript has addressed the concerns raised in my previous review, and I support publication in Nature Communications.

COMMENTS TO AUTHOR:

Reviewer 1: I already reviewed the originally submitted paper by Mao & Walsh, which is an interesting contribution in the context of formal C–H bond activation –via a supposed transition metal-free deprotonation– of challenging low-acidity substrates (toluene and derivatives; pKa ~ 43). Regarding the revised version of this manuscript, I am fairly pleased with the efforts of the authors although there are still a few points to be addressed.

Response: We appreciate the reviewer’s support and her/his recognition that C–H bond activation and functionalization of toluene is challenging and of “high novelty”.

Reviewer 1 continues: The background is now (almost) complete and accurate, although a few adjustments are required in terms of order, referencing, and visibility of the chemistries displayed in Scheme 2. I believe that the benzylic C–H bond activations “D–E–F” are more accurately to be shown in a chronological order; Schneider (2017), Kobayashi (2017, plus concept article in 2018), Guan (2018); means: D & E should be swapped (likewise the corresponding refs), and the text must be adjusted accordingly.

Response: We have swapped D and E. The refs have also been changed accordingly (ref²⁵ and ref^{26,27}).

Reviewer 1 continues: In the paper’s context (benzylic substrates), ref 26 is not required because it is not on benzylic C–H bond activation, but rather on “ordinary” catalytic enolate-type chemistry; also, this paper is cited already in the concept article – thus, please remove ref 26 (otherwise, many others would need to be added!). While it is appreciated that refs 27 and 31 (published very recently) have been included in the References, I would like to see these briefly contextualized in the text (not as a Scheme); ref 31 is not “competitive” in terms of pKa values anyway, but ref 27 is very much “competitive” here, and it is not only catalytic but also includes the addition to alkenes as well as a preliminary asymmetric version (imine addition). Finally, please add the term “catalytic” to both Schneider’s and Guan’s works (legend of Scheme 2); these catalyses must be distinguished from the stoichiometric studies (not only in the artwork).

Response: We have removed ref²⁶ and added “During the revision process, Brønsted

bases catalyzed benzylic C–H bond functionalizations of toluenes and diarylmethanes were reported by Kobayashi²⁹ and Guan³⁰, respectively” in the revised manuscript. The term “catalytic” has been added in the legend.

Reviewer 1 continues: From a general point of view, the argumentation (in the rebuttal) about the lengthy imine preparation in Kobayashi’s work is not so relevant because their reaction is catalytic in metal amide –which should be more appreciated– and it works also with alkenes (using a tridentate amine ligand as additive); a particularly difficult transformation. Anyway, I still think that the mediator system in this work –vs. Kobayashi’s work– is somewhat different, which gives credit to the Mao & Walsh: Kobayashi used a lithium amide activated by a potassium alkoxide (Lewis base), whereas Mao & Walsh used a sodium amide and a cesium salt (pi-Lewis acid), the latter of which activates the benzylic substrates. Means: the activation mode seems to be quite distinct. In this manuscript, a TON of 2 was achieved (Table1, entry 17), which is not great but shows that catalysis should be in principle feasible. It is fair to say that the various (other) points raised earlier by myself and by the other reviewers have been addressed satisfactorily. For instance, the distinction to the AZ work is not great but sufficient. The delivered control experiments and explanations are fine (T1 & SI); likewise, the displayed scope (T2 & T3) is acceptable; I am still not convinced regarding the utility of S4 though.

Response: We are happy that the reviewer is supportive of our changes to the original manuscript. We chose to keep Scheme 4, because scalability is very important for a novel reaction. Furthermore, gram scale reaction could demonstrate its potential applications. We anticipate the methods for synthesis of these bioactive compounds (indolines and 1,2-diphenylethylamine derivatives) could attract the interests from medicinal chemists.

Reviewer 1 continues: The requested SI adjustments are fine as well. The displayed mechanism (S3) is straightforward; one detail –regarding the structure of key intermediate C– must be checked/adjusted though prior to publication: is it really to be represented as a sigma-complex, or isn’t it rather a pi-complex (like known for e.g. benzyl–Pd species)? Thus, please verify in the literature the eta¹ vs. eta³ complexation mode for this alkali metal–benzyl intermediate. It should be known in an inorganic context. It may be best to draw both species in an equilibrium anyway (?)..

Response: We have drawn both species in an equilibrium as below and added one ref accordingly (*J. Am. Chem. Soc.* **1994**, *116*, 528-536).

Reviewer 1 continues: Conclusion: Because of the highly challenging pKa value of the substrates, the distinct activation mode (vs. Kobayashi 2018), and the catalytic proof-of-principle (while not great yet), I would say that I can recommend publication in *Nat. Commun.* if the mentioned points above are addressed in full.

Response: We appreciate the reviewer's support and recommendation for publication of the revised manuscript.

Reviewer 2 writes: This revised manuscript has addressed the concerns raised in my previous review, and I support publication in *Nature Communications*.

Response: We appreciate the reviewer's support.

We thank the reviewers for their helpful suggestion. With the changes we have made, we hope that the manuscript is now acceptable for publication in *Nature Communications*.